# Genome-Wide Study of Conidiation-Related Genes in the Aphid-Obligate Fungal Pathogen *Conidiobolus obscurus* (Entomophthoromycotina)

**DOI:** 10.3390/jof8040389

**Published:** 2022-04-12

**Authors:** Lvhao Zhang, Tian Yang, Wangyin Yu, Xiaojun Wang, Xiang Zhou, Xudong Zhou

**Affiliations:** State Key Laboratory of Subtropical Silviculture, School of Forestry and Biotechnology, Zhejiang A&F University, Hangzhou 311300, China; 2020102081016@stu.zafuedu.cn (L.Z.); 2021102032004@stu.zafu.edu.cn (T.Y.); yuwangyin98@gmail.com (W.Y.); 2020102082013@stu.zafuedu.cn (X.W.)

**Keywords:** mycopathogen, fungal genome, fungal virulence, genomic and transcriptomic profiling, insect pathogenic fungi

## Abstract

Fungi in the Entomophthorales order can cause insect disease and epizootics in nature, contributing to biological pest control in agriculture and forestry. Most Entomophthorales have narrow host ranges, limited to the arthropod family level; however, rare genomic information about host-specific fungi has been reported. Conidiation is crucial for entomopathogenic fungi to explore insect resources owing to the important roles of conidia in the infection cycle, such as dispersal, adhesion, germination, and penetration into the host hemocoel. In this study, we analyzed the whole genome sequence of the aphid-obligate pathogen *Conidiobolus obscurus* strain ARSEF 7217 (Entomophthoromycotina), using Nanopore technology from Biomarker Technologies (Beijing, China). The genome size was 37.6 Mb, and encoded 10,262 predicted genes, wherein 21.3% genes were putatively associated to the pathogen–host interaction. In particular, the serine protease repertoire in *C. obscurus* exhibited expansions in the trypsin and subtilisin classes, which play vital roles in the fungus’ pathogenicity. Differentially expressed transcriptomic patterns were analyzed in three conidiation stages (pre-conidiation, emerging conidiation, and post-conidiation), and 2915 differentially expressed genes were found to be associated with the conidiation process. Furthermore, a weighted gene co-expression network analysis showed that 772 hub genes in conidiation are mainly involved in insect cuticular component degradation, cell wall/membrane biosynthesis, MAPK signaling pathway, and transcription regulation. Our findings of the genomic and transcriptomic features of *C. obscurus* help reveal the molecular mechanism of the Entomophthorales pathogenicity, which will contribute to improving fungal applications in pest control.

## 1. Introduction

Filamentous fungal species undergo asexual reproduction in their life cycle by the formation and release of asexual spores from aerial hyphae [1,2]. Conidiospores (conidia) are the main asexual fungal spores that favor fungal spread, survival, and colonization in new habitats [3,4]. Pathogenic fungi also use conidia as infectious propagules to recognize and infect host plants and animals [5,6]. Entomopathogenic fungi represent over 1000 species, mainly in the orders Hypocreales (ca. 600 species belonging to the filamentous Ascomycetes) and Entomophthorales (ca. 300 species belonging to the basal fungi in Zoopagomycota), are widely distributed in the environment, and regulate host insect populations by causing mycosis and epizootics [5,7]. These entomopathogens develop millions of differently shaped clonal conidia initiating their infection cycles (conidial adhesion, infection structure differentiation, detoxification, insect hemocoel adaptation, and host nutrient deprivation) [6,8]. Most Entomophthorales are host-specific and have unique ecological interactions with their hosts compared to the Hypocreales, which have wide host ranges. The fungal entomopathogens secrete subtilisin-like serine proteases to degrade chitin-associated proteins in the insect procuticle, and the kinds of subtilisin-like proteases secreted are regarded as being associated to their host ranges [9,10]. For breaking through the external layer of the insect cuticle (hydrocarbon and lipid-rich epicuticle), cytochrome P450 monooxygenases (P450s, CYP) and lipases are also secreted [11,12]. Conidia-based formulations of Hypocreales for insect pest control have been applied, but the formulation and application of Entomophthorales remain stagnant owing to their unique conidiation process and conidial characteristics [13]. The Entomophthorales can actively eject primary conidia from conidiophores (conidiation) after their vegetative (pre-conidiation) stage, and shed the energy-exhausted mycelial residues (post-conidiation). The germination pathways of Entomophthorales conidia demonstrate the fungal responses to different environments. After landing on non-host surfaces, secondary and tertiary conidia are ejected from the primary and secondary conidia, respectively, through the extended germ tubes to increase the chances of encountering hosts [3]. Once the host is recognized, the conidia germinate into infectious germ tubes and penetrate the host cuticles for several hours under warm and moist conditions [14,15]. After the infected host dies, the fungi start conidiation on the cadavers to enter the new infection cycle. Thus, the Entomophthorales conidiation facilitates the spread of pathogens in host populations and epizootic outbreaks, ultimately causing the collapse of pest cohorts.

The air-borne conidia are generally dormant when they detach from conidiophores. The ability of the conidia of entomopathogens to rapidly settle on host or non-host surfaces probably relies on the conidiation process, such as mucinous fluid wrapping primary conidia for adhesion [5]. Conidiation is a complex biological process, including cellular component biosynthesis, nutrient allocation, and storage, and is regulated by environmental factors such as light and nutrient limitation [2]. Variation in gene expression is the basis for morphological and physiological heterogeneity among mycelia and conidia [16,17]. Profiling gene expression patterns during the conidiation period favors understanding of the infection mechanism and the specific roles of conidia of entomopathogens.

Genomic studies provide new insights into phylogenetic evolution and biological process of entomopathogens [18]. However, few genomic studies of Entomophthoromycotina (Zoopagomycota) have been reported, except on members of the paraphyletic *Conidiobolus* genus, which includes saprophytes, and pathogens of insect, animals, and humans [5,18]. These studies investigated the genes encoding plant cell wall digestion enzymes, genes encoding P450s, and other saprotrophy-related genes of *Conidiobolus* spp., proving that the diversity of those genes contributes to the fungal ecological niches [19,20,21]. In this study, we report the genome sequence of the aphid-obligate fungal pathogen *C. obscurus* ARSEF 7217, and screen the conidiation-related genes that are involved in the pre-conidiation (pre-C), emerging conidiation (C), and post-conidiation (post-C) stages based on transcriptomic analysis. This study will enhance our understanding of the molecular characteristics of entomopathogenic fungi and will aid in exploring the functional genes related to conidiation and pathogenicity, with the aim of improving their application in agroforestry pest control.

## 2. Materials and Methods

### 2.1. Fungal Culture

*C. obscurus* 7217 was obtained from the United States Department of Agriculture-Agricultural Research Service Collection of Entomopathogenic Fungal Cultures (USDA-ARSEF, Ithaca, NY, USA) and stored at −80 °C [22]. The isolate was cultured on rich Sabouraud dextrose agar plus yeast extract (40 g L^−1^ dextrose, 10 g L^−1^ peptone, 10 g L^−1^ yeast extract, and 15 g L^−1^ agar) for 4 days in Petri dishes at 24 ± 1 °C with a 12:12 h light: dark photoperiod. Culture pieces were mashed and transferred into 50 mL liquid media (150 mL flask) and incubated in a shaker at 120 rpm and 24 ± 1 °C for 3 d. Fresh mycelia (pre-C stage) were collected using a 0.2 µm filter, and evenly poured into a 90 mm Petri dish to form a mycelial mat while removing any excess water using sterile paper. Moist conditions (100% relative humidity) were maintained for 12 h at 24 °C to form the C stage. The mycelial mats maintained for 72 h at the same temperature and humidity represented the post-C stage.

### 2.2. Genome Sequencing, Assembly, and Annotation

Fresh liquid-cultured mycelium was collected and ground into a fine powder in liquid nitrogen for DNA extraction. Total genomic DNA was extracted following the protocol set forth in the TakaRa universal genomic DNA extraction kit (Tokyo, Japan) for sequencing. Genome sequencing was carried out by Biomarker Technologies (Beijing, China) on a nanopore sequencing platform using the standard protocol provided by Oxford Nanopore Technologies [23]. Raw data were deposited in the CNGB Sequence Archive (CNSA) of the China National GenBank Database (CNGBdb, https://db.cngb.org/ accessed on 1 January 2022) under the accession number CNP0001555 [24]. After filtering low quality reads and short reads (<2000 bp), nanopore reads longer than 5 kb were selected for error correction using Canu v1.7.1 [25]. Clean reads obtained from Canu were assembled using wtdbg [26]. The Benchmarking Universal Single-Copy Orthologs (BUSCO) tool v2.0 was used to assess completeness of the final assembly using the fungi_odb9 dataset [27].

Repeat sequences were predicted using RepeatMasker 4.06 [28]. tRNAscan-SE and Infernal 1.1 software were used for tRNA and rRNA identification, respectively [29,30]. Structural annotation was performed using Augustus v2.4 as gene predictor [31]. GeMoMa v1.3.1 was used for homolog-based gene prediction with the genome sequence of *C. coronatus* [20,32]. RNA-seq data generated in this study (CNGBdb accession number: CNP0002115) were also used as supporting gene evidence during the annotation process.

The functions of the putative protein-coding genes were annotated using the basic local alignment search tool (BLASTx) with an of E-value < 10^−5^. Public databases of NCBI non-redundant protein sequences (NR), NCBI eukaryotic orthologous groups of proteins (KOG), Swiss-Prot, Pfam, Gene Ontology (GO), Kyoto Encyclopedia of Genes and Genomes (KEGG), Fungal Transcription Factor Database, and Pathogen–Host Interaction database (PHI, http://www.phi-base.org accessed on 9 February 2021) platforms were used. Secretory proteins were screened based on their structures using the signal peptide predicted by SignalP 5.0 without the membrane-spanning domain [33]. CAZymes were identified as previously described [21].

### 2.3. RNA Extraction and Transcript Assembly

Three biological replicates were sampled from each of the three stages: pre-C, C, and post-C to screen for conidiation-related genes in *C. obscurus*. Total RNA was extracted using the RNAiso Plus kit (TaKaRa, Tokyo, Japan), and its concentration was measured using a NanoDrop 2000 spectrophotometer (Thermo Fisher Scientific, New York, NY, USA). RNA degradation and contamination (especially DNA contamination) were monitored on 1% agarose gels and samples were sent to Biomarker Technologies (Beijing, China) for transcript sequencing. Ribosomal RNA was removed from total RNA using a Ribo-Zero rRNA removal kit (Epicenter, Madison, WI, USA). A total of 1.5 µg rRNA-free RNA per sample was used to generate a sequencing library using the NEBNext Ultra Directional RNA library prep kit for Illumina (New England BioLabs, Boston, MA, USA) on an Illumina HiSeq 4000 platform (BGI, Beijing, China) according to the sequencing pipeline (Appendix A). The adapter sequences were trimmed to remove low-quality sequences, and the high-quality clean reads were aligned to the genome of *C. obscurus* 7217 using hierarchical indexing for spliced alignment of transcripts (HISAT2) [34]. Reads with no more than two mismatches from each sample were used to generate the transcripts using StringTie software 1.3.1 [35]. The workflow of the HISAT-StringTie analysis is shown in Appendix A.

### 2.4. Quantification of Transcripts

Transcript expression was quantified as fragments per kilobase of transcript per million mapped reads (FPKM) using StringTie 1.3.1 [35]. Differential expression analysis of the transcripts between the three libraries was performed using the DESeq R package 1.10.1 [36]. The resulting *p*-values were adjusted using the Benjamini–Hochberg method (multiple hypothesis test) for controlling the false discovery rate (FDR). Transcripts with an adjusted *p*-value of less than 0.01 and an absolute value of log2 (fold change, FC) of more than 1 were designated as differentially expressed [34]. Quantitative real-time polymerase chain reaction (qRT-PCR) was performed using designed primers (Appendix A) to evaluate the relative expression levels of the selected genes for further validation of the findings.

### 2.5. Co-Expression Network Construction

The distinct hub genes of conidiation were established using a weighted gene co-expression network analysis (WGCNA v1.69 package on the BMKCloud platform) [37]. Gene expression values were imported into WGCNA to construct co-expression modules using the automatic network construction function block-wise modules with the default settings. The expression levels of the differentially expressed genes (DEGs) were log-transformed using log2 (FPKM + 1). Pearson’s correlation coefficient was used to determine the co-expression relationship between each pair of genes. The WGCNA network was constructed with a soft thresholding power of β = 17, a minimum module size of 30 genes, the TOM-Type was unsigned, and the merge cut height was 0.25. The module–trait relationship was used to differentiate the hub genes among the conidiation stages.

### 2.6. Phylogenetic Analysis

The MEGAX software suite was used to determine the phylogenetic relationships of the genes encoding the serine proteases [38]. The phylogenetic tree was constructed using the maximum likelihood method based on the Poisson correction model, with 500 bootstrap replicates. The protein sequences and information were obtained from the assembled genome of *C. obscurus* 7217 (CNP0001555).

### 2.7. Comparative Genomic Analysis

The wide-host-range generalist *Conidiobolus coronatus* (GenBank assembly accession: GCA_001566745.1) [20] was used for comparison with genomic information obtained for the assembled genome of *C. obscurus* 7217 (CNP0001555). The gene family was constructed and compared using OrthoMCL [39].

## 3. Results

### 3.1. Global Characteristics of the C. obscurus Genome

Nanopore sequencing generated a total of 11.9 Gb clean reads, and the net length of *C. obscurus* genome sequence was 37.6 Mb, with the assembly consisting of 167 scaffolds and a 26.46% G+C content (Table 1 and Appendix A). In total, 10,262 protein-encoding genes were predicted, of which 84.08% were homolog- and RNA-seq-based predictions. A total of 587 non-coding RNAs (ncRNAs) in 67 families were identified, including 526 tRNAs, 8 rRNAs, and 49 other ncRNAs. The total length of repeat sequences was 6.6 Mb, representing 17.44% of the genomic length (Appendix A).

A total of 8566 of the 10,262 genes (83.5%) was mapped to different databases to explore the functions (Appendix A), wherein 2375 genes were annotated in GO, 4321 genes were annotated in KEGG, and 6465 genes were annotated in KOG. The GO-annotated genes (Figure 1) were categorized into three major classes, molecular function (11 sub-classes), cellular component (12 sub-classes), and biological process (18 sub-classes). The top five functional entries included catalytic activity (1339 genes), metabolic process (1074 genes), cellular process (1028 genes), binding (906 genes), and cell part (703 genes). The KEGG-annotated genes (Figure 2) were categorized into four major classes, metabolism (25 sub-classes), genetic information processing (17 sub-classes), cellular processes (5 sub-classes), and environmental information processing (3 sub-classes). The top five functional entries included ribosome (150 genes), protein processing in endoplasmic reticulum (133 genes), RNA transport (123 genes), biosynthesis of amino acids (123 genes), and oxidative phosphorylation (112 genes). The KOG-annotated gene functions (Figure 3) were centered on posttranslational modification, protein turnover, chaperones (789 genes), and signal transduction mechanisms (564 genes).

### 3.2. Annotation in PHI and CAZy Databases

PHI-base is a database that catalogues the experimentally verified pathogenicity and effector genes from pathogens of animal, plant, fungal, and insect hosts. In the *C. obscurus* genome, 2888 genes were associated to the PHI database, 75.9% of which were putatively involved in the fungal pathogenicity. Results revealed that the maximum (1306) number of matched genes was related to the functional class of “reduced virulence”, while 695 genes were identified as “unaffected pathogenicity” (Appendix A). Based on KOG classes, many of these PHI-associated genes function in signal transduction mechanisms (224 genes, representing 39.7% in T class), post-translational modification, protein turnover, chaperones (219, 27.8% in O class), lipid transport and metabolism (186, 42.1% in I class), and secondary metabolites biosynthesis, transport, and catabolism (171, 72.2% in Q class) as shown in Figure 3.

The CAZy database describes the families of carbohydrate-active enzymes (CAZymes) that degrade, modify, or create glycosidic bonds. The members of CAZymes in *C. obscurus* based on the CAZy database contained 5 main modules, including 132 glycoside hydrolases (GH, in 20 families), 122 glycosyl transferases (GT, in 22 families), 74 carbohydrate esterases (CE, in 8 families), 38 carbohydrate-binding modules (CBM, in 8 families), and 1 polysaccharide lyase (PL) (Figure 4, and Appendix A).

In the genome, 125 genes were associated with cytochrome P450s. Many (28) of the genes (P450-DIT2, CYP56) putatively involved in conidial wall maturation by catalyzing the oxidation of tyrosine residues in the formation of LL-dityrosine (a precursor of the cell wall) were found. In total, 11 CYP52-coding genes were found, which, putatively together with NADPH cytochrome P450 (CYP505, 6 genes), were involved in the first step in the assimilation of alkanes and fatty acids (the main components in the host epicuticle). Others (Appendix A) were predicted to encode enzymes involved in detoxification and secondary metabolite production.

### 3.3. Abundance of Serine Protease-Encoding Genes in C. obscurus

Based on Pfam annotation, the putative peptidase-encoding genes of *C. obscurus* (328 genes) consisted of serine proteases, metallopeptidases, and cysteine peptidases. Ten serine protease families, including trypsin (S1A, 67 genes), subtilisin (S8, 52), prolyl oligopeptidase (S9, 10), carboxypeptidase Y (S10, 2), Lon protease (S16, 2), signal peptidase I (S26, 4), acid prolyl endopeptidase (S28, 12), prolyl aminopeptidase (S33, 11), rhomboid (S54, 4), and nucleoporin (S59, 2) were identified in *C. obscurus* (Figure 5).

### 3.4. Comparative Genomic Analysis

There are 3538 common gene families between *C. obscurus* and *C. coronatus* (Figure 6A), and the difference in the unique gene families probably contributes to the distinct host ranges of these two fungal pathogens. Based on Pfam annotation, the unique functional genes in *C. obscurus* are enriched in trypsin, glycosyl hydrolase, peptidase, protein kinase, multicopper oxidase, P450s, transcription factors (fungal Zn(2)-Cys(6) binuclear cluster domain and homeobox domain), and RNA recognition motif (Figure 6B).

### 3.5. Gene Differential Expression in the three Conidiation Stages

The mycelia before, during, and after conidiation showed obvious morphological differences by changing from full and round hyphae to spherical conidia formation in hyphal tips, and to shriveled hyphae (Figure 7). Approximately 90.7 Gb of clean reads were generated from the nine libraries of *C. obscurus* mycelia in the three stages, and 62.4–84.3% reads were mapped to the reference genome (Appendix A). Transcriptome analysis identified 3091 DEGs (1879 upregulated and 1212 downregulated) in C vs. pre-C, and 3235 DEGs (1954 upregulated and 1281 downregulated) in C vs. post-C based on transcripts with a FC ≥ 2 and FDR ≤ 0.01 (Appendix A).

The functions of the DEGs in C vs. pre-C were concentrated in amino sugar and nucleotide sugar metabolism, biosynthesis of amino acids, biosynthesis of antibiotics, carbon metabolism, and starch and sucrose metabolism based on KEGG functional enrichment analysis (Figure 8A). Similarly, the functions of the DEGs in C vs. post-C were concentrated in biosynthesis of amino acid, 2-oxocarboxylic acid metabolism, biosynthesis of antibiotic, and lysine biosynthesis (Figure 8B).

Genes related to conidiation in *C. obscurus* were screened based on the criteria of being upregulated in the C stage versus either the pre-C or post-C stages. In total, 2915 DEGs were putatively related to conidiation wherein 918 DEGs were upregulated in both C stage vs. pre-C stage and C stage vs. post-C stage (Appendix A). The most upregulated DEGs during the conidia-emerging phase included those encoding glycosylphosphatidylinositol (GPI)-anchored protein, tryptophan-rich sensory protein (TspO), polysaccharide deacetylase, lipase, cytochrome P450, homeoprotein, trypsin, subtilisin-like serine proteinase, glycosyl hydrolase, and AphC/TSA antioxidant enzyme (Table 2).

There were only 65 CAZyme-encoding genes upregulated in conidiation, including 14 genes in GH18 family (endochitinase) (Figure 4). A total of 375 DEGs putatively encoding transcription factors in 25 transcription factor families were upregulated in conidiation (Appendix A). The relative expression level of the selected genes from the three conidiation stages revealed by qRT-PCR corroborated the data obtained from analyzing the transcriptome (Appendix A).

### 3.6. Co-Expression Network Analysis of Conidiation-Related Genes

A WGCNA across all nine samples elucidated the conidiation-related genes. Highly interconnected genes were grouped into three different modules (black, brown, and turquoise) (Figure 9). The black module containing 772 DEGs showed the strongest correlation with conidiation including 580 DEGs upregulated in both C vs. pre-C and C vs. post-C. Most of these DEGs were grouped into GO terms related to cell parts, catalytic activity, and metabolic processes (Appendix A) and enriched in KEGG pathways involved in oxidative phosphorylation, ribosomes, and MAPK signaling (Appendix A). The KOG terms were concentrated in “post-translational modification, protein turnover, chaperones”, “carbohydrate transport and metabolism”, and “cell wall/membrane/envelope biogenesis” (Appendix A).

DEGs in the black module annotated by the PHI database showed that putative virulence factors were mainly associated with diverse secretory proteins, including subtilisin-like serine proteases, trypsin-like serine proteases, lipases, and chitinases (Appendix A). A heat map using hierarchical clustering analysis of those genes showed that the nine samples localized into three clusters at each of the pre-C, C, and post-C stages wherein C stage DEGs had larger FPKM values than those in the other two stages (Figure 10).

## 4. Discussion

The subphylum Entomophthoromycotina (Zoopagomycota) includes arthropod-pathogenic fungi, saprophytes, and human pathogens and boasts a large range of genome sizes, with the largest one at 8000 Mb [18]. In our study, the genome size of the aphid-obligate pathogen *C. obscurus* ranges between the 34.4 Mb of the saprobiotic *Conidiobolus heterosporus* [21] and 39.9 Mb of *C. coronatus* with a broad host range [20]. The G+C content of *C. obscurus* is closer to *C. coronatus* (27.7%) than to *C. heterosporus* (36.8%), consistent with the low G+C content (<40%) usually observed in the basal fungi of Zoopagomycota [21]. Genome duplication and gene family expansions contribute to speciation and host specialization [18]. In the comparison, the functions of the unique genes and gene families in the *C. obscurus* genome were concentrated on host cuticular component degradation and transcription regulation (Figure 6), implying the specialization in host-pathogen interaction. A total of 23.1% of the predicted genes of *C. obscurus* were putatively related to pathogenicity based on PHI annotation, higher than an average of 15% of total proteins as determinants of fungal virulence [6]. Among these, 806 genes were upregulated during conidiation, demonstrating the intricate association of conidiation and fungal pathogenicity.

There are significant functional differences in the CAZymes family members, although the number of CAZymes-encoding genes are comparable in *C. obscurus* (392) and *C. heterosporus* (394). The genes in CE12 (pectin acetylesterases) and CBM12 (chitin-binding) families are abundant for the saprobiotic lifestyle of *C. heterosporus* to degrade leaf molds and plant detritus [21]. In our study, the dominant genes belonged to GH18, GT1, and CE10 families, consistent with the differential nutrient-acquisition strategies, and only 16.6% of CAZymes-encoding genes are involved in conidiation. In entomopathogens, P450s have multiple functions, facilitating penetrating through the host epicuticle, detoxification, and increasing virulence by toxic secondary metabolite production [11,12]. In the *C. obscurus* genome, the number of P450s-encoding genes (125) are less than those (142) in *C. coronatus* [20], while 13.6% were upregulated in conidiation (Appendix A), probably contributing to the fungus’ pathogenicity.

The conidia of fungal entomopathogens play multiple roles in dispersal, survival, and exploitation of nutritious resources in new habitats or hosts [5]. Fungal conidiation features morphological changes (tip growth) of conidiophores expanding in surface area and cell volume. This involves degradation, biosynthesis, and assembly (cross-linking) of conidial wall and membrane components such as glucan, chitin, mannans, glycoproteins, and ergosterol [40]. For example, glycosyl hydrolase and 1,3-β-glucan synthase are involved in cell wall remodeling and regulating cell wall thickness [41]. In our study, many genes encoding glycosidases, glucan synthases, and chitin synthases were upregulated (Appendix A). The gene encoding GPI-anchored protein, which is an abundant constituent of the eukaryotic cell surface and played a pathogenic role in fungal infection, was markedly upregulated during conidiation in this study [42]. This explains the dynamic transport of amino acids, lipids, carbohydrates, and iron in the conidiation process.

The Entomophthorales conidia are coated with a thick film of mucinous fluid that contain secreted enzymes, facilitating their adhesion and subsequent infection of the hosts [43]. Many omics studies on the entomopathogenicity in Entomophthoromycotina revealed core fungal virulence factors are secreted enzymes, including subtilisin- and trypsin-like serine proteases, chitinases, and lipases [14,44,45]. Most of the sequenced fungal lineages encode a set of 13–16 serine protease families that perform a variety of functions [46]. *C. obscurus* has only 10 families of serine proteases but exhibits the gene expansions of trypsin and subtilisin families. Diversifying the subtilisin-like proteases is an advanced evolutionary tactic to enhance pathogenicity in the wide-spectrum entomopathogenic fungi, such as *Beauveria bassiana* and *Metarhizium anisopliae* [10,47]. The proteinase-K-like fungal S8 is the principal subtilisin-like serine proteases in Entomophthoromycotina that exhibits remarkably high host-specificity for insects in pathogenic fungi, consistent with *C. obscurus* [9]. Additionally, the gene number (52) of these subtilisin-like proteases is significantly higher in *C. obscurus* than in *C. coronatus* (36), *C. thromboides* (18), and *Entomophthora muscae* (22). Most genes encoding the S1 family (chymotrypsin) in *C. obscurus* are also in the S1A subfamily (trypsin) for extracellular degradation. Many subtilisin (S8)- and trypsin (S1A)-encoding genes were upregulated during conidiation, depicting their precise correlation with the fungal pathogenicity.

Conidia are aerial-born spores that have adapted to harsh conditions such as oxidative stress and UV radiation. Several DEGs related to environmental stress responses during conidiation were observed in this study. For example, five putative genes encoding peroxiredoxin in AhpC/TSA family may participate in the defense against oxidative damage response, probably comprising a strong antioxidant machinery to collapse the host immune system, favoring the fungus exploring the nutrients in the host hemolymph [48]. Nine genes encoding tryptophan-rich sensory protein (TSPO), the translocator proteins which play pivotal role in transporting cholesterol inside mitochondria, were substantially upregulated. These genes may be involved in the cellular response to environmental changes in oxygen and light conditions [49]. Two upregulated DEGs (EVM0003905 and EVM0001948) homologous to ATP-dependent DNA helicase may be involved in DNA repair following UV-induced damage [50].

The gene expression patterns involved in conidiation require transcriptional regulation by regulatory proteins and components of signal transduction pathways [2]. Previous studies have revealed the involvements of homeobox genes in conidiation, which are evolutionarily conserved in eukaryotes [51,52]. This study revealed various genes encoding diverse families of transcription factors that were upregulated during conidiation. Several such genes encode homeoproteins (Appendix A). Moreover, we also observed 15 genes encoding the components of MAPK signaling pathway that were significantly upregulated, suggesting an essential role of MAPK pathway in the conidiation process [53]. For example, EVM0001653 is putatively related to the plasma membrane osmosensors that activate the high osmolarity glycerol (HOG) MAPK signaling pathway, which may favor the fungal adaptation to the osmotic pressure of the host hemolymph and appressorium development [53,54]. Moreover, many transcription factors are encoded by the unique genes in the *C. obscurus* genome (Figure 6), implying that the fungal conidiation process is suitable for its unique ecological niche.

## 5. Conclusions

Our present study provides valuable information about the genomic features of the aphid-obligate pathogen *C. obscurus* and enhances understanding of the unique characteristics of Entomophthoromycotina, considering that most species in this subphylum are obligate pathogens of arthropods and there are few reported genomes. Based on the theoretical support of our whole-genome sequencing, the transcriptomic profiling provides new insights into the molecular mechanism of the conidiation process, showing that the abundance of transcripts with diverse functions is the basis for the multiple roles of conidia in mycosis. These results also provided a basis for further understanding the gene network for the infection cycle of entomopathogenic fungi and reference information for studying other related fungi, contributing to the further application of Entomophthoromycotina in agricultural and forestry pest control.

## Figures and Tables

**Figure 1 jof-08-00389-f001:**
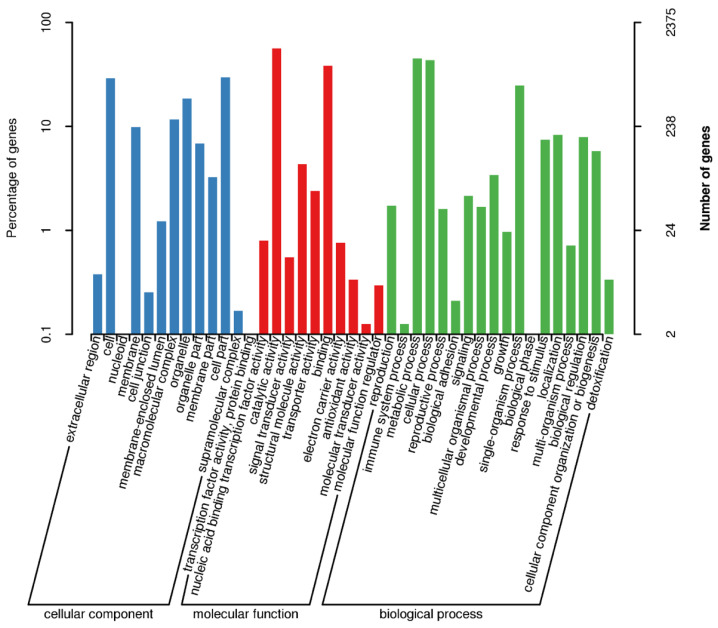
GO functional annotation of the *C. obscurus* genome. The *x*-axis shows in different colors the protein-coding genes divided into three major categories. The left *y*-axis represents the percent of genes (%) while the number of the genes are represented by the right *y*-axis.

**Figure 2 jof-08-00389-f002:**
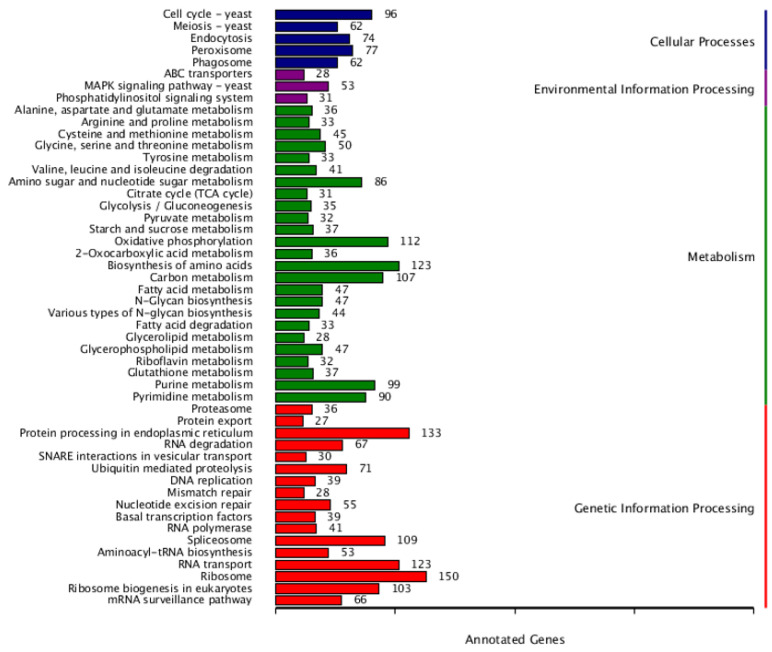
The KEGG functional annotation of *C. obscurus* genome. The major classes (distinct color) with their names and the number of genes from the concerned sub-class divisions are represented.

**Figure 3 jof-08-00389-f003:**
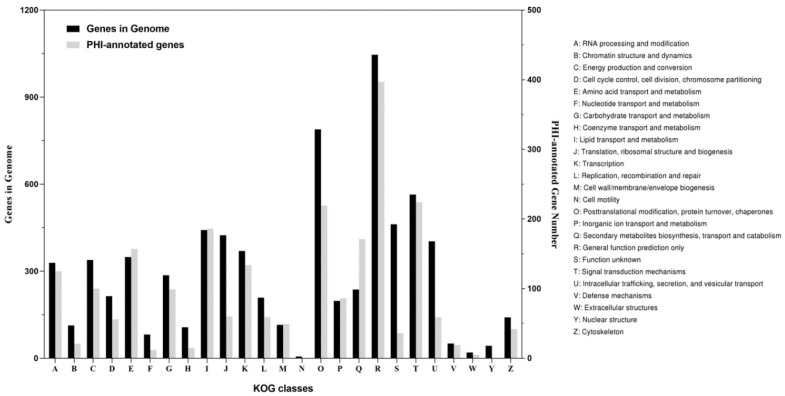
The KOG classes in the *C. obscurus* genome are associated with records in the PHI database. The *x*-axis represents the classes of genes and left *y*-axis represents the number of genes while the right *y*-axis represents the number of the PHI-annotated genes.

**Figure 4 jof-08-00389-f004:**
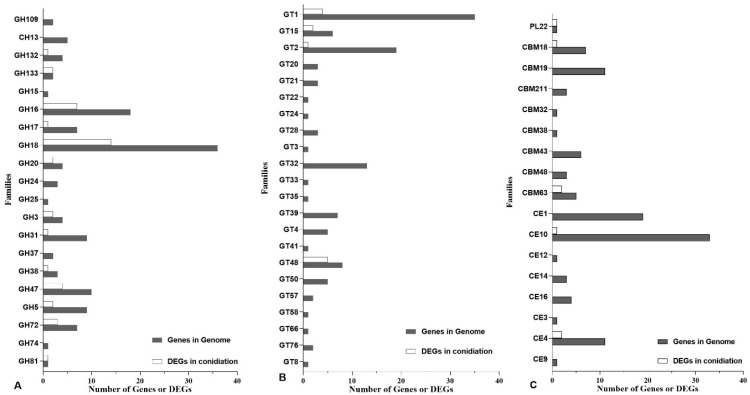
Gene numbers of CAZyme-encoding genes in the *C. obscurus* genome and the related differentially expressed genes (DEGs) in conidiation. (**A**) glycoside hydrolase (GH) family; (**B**) glycosyl transferase (GT) families; (**C**) carbohydrate esterase (CE), carbohydrate-binding module (CBM), and polysaccharide lyase (PL) families. CAZyme families were identified with the CAZyme database (http://www.cazy.org, accessed on 14 March 2022).

**Figure 5 jof-08-00389-f005:**
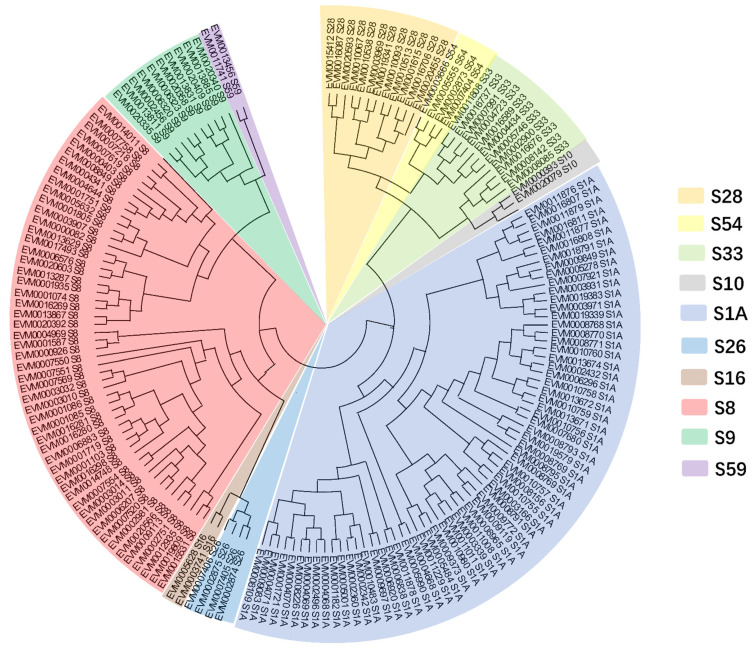
Phylogenetic tree for the serine proteases of *C. obscurus* based on the maximum likelihood method using MEGAX. Serine protease families: S1A (trypsin), S8 (subtilisin), S9 (prolyl oligopeptidase), S10 (carboxypeptidase Y), S16 (Lon pro-tease), S26 (signal peptidase I), S28 (acid prolyl endopeptidase), S33 (prolyl aminopeptidase), S54 (rhomboid), and S59 (nucleoporin).

**Figure 6 jof-08-00389-f006:**
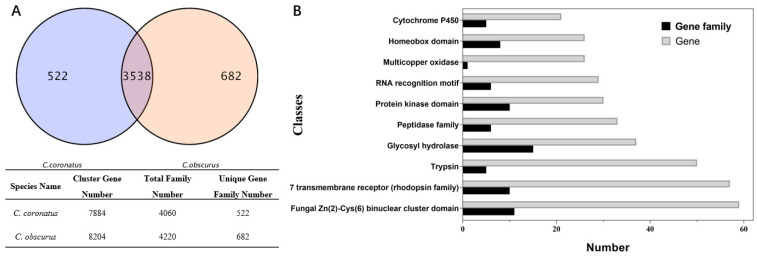
The comparative genomic analysis of *C. obscurus* and *C. coronatus* using OrthoMCL. (**A**) The common and unique gene family (**B**) The top 10 Pfam-annotated unique genes in the *C. obscurus* genome.

**Figure 7 jof-08-00389-f007:**
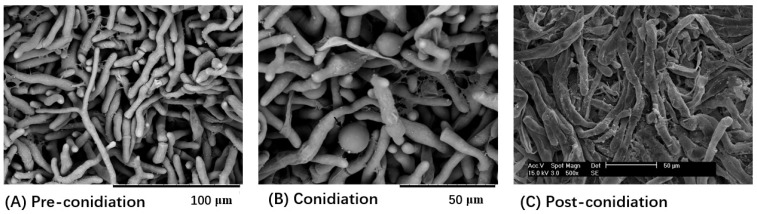
Scanning electron microscopy images of *C. obscurus* mycelium before, during, and after conidiation. For observing the morphological characteristics of *C. obscurus* in different stages of conidiation, the mycelial samples were collected. After fixation, dehydration, and platinum ion plating, the specimens of the mycelial mats of three conidiation stages were separately observed with Philips XL30-ESEM environmental scanning electron microscope (SEM). (**A**) Fresh liquid-cultured mycelia in the pre-conidiation (pre-C) stage; (**B**) Conidiation mycelia with global conidia on the tips; (**C**) Shriveled mycelia post-conidiation (post-C).

**Figure 8 jof-08-00389-f008:**
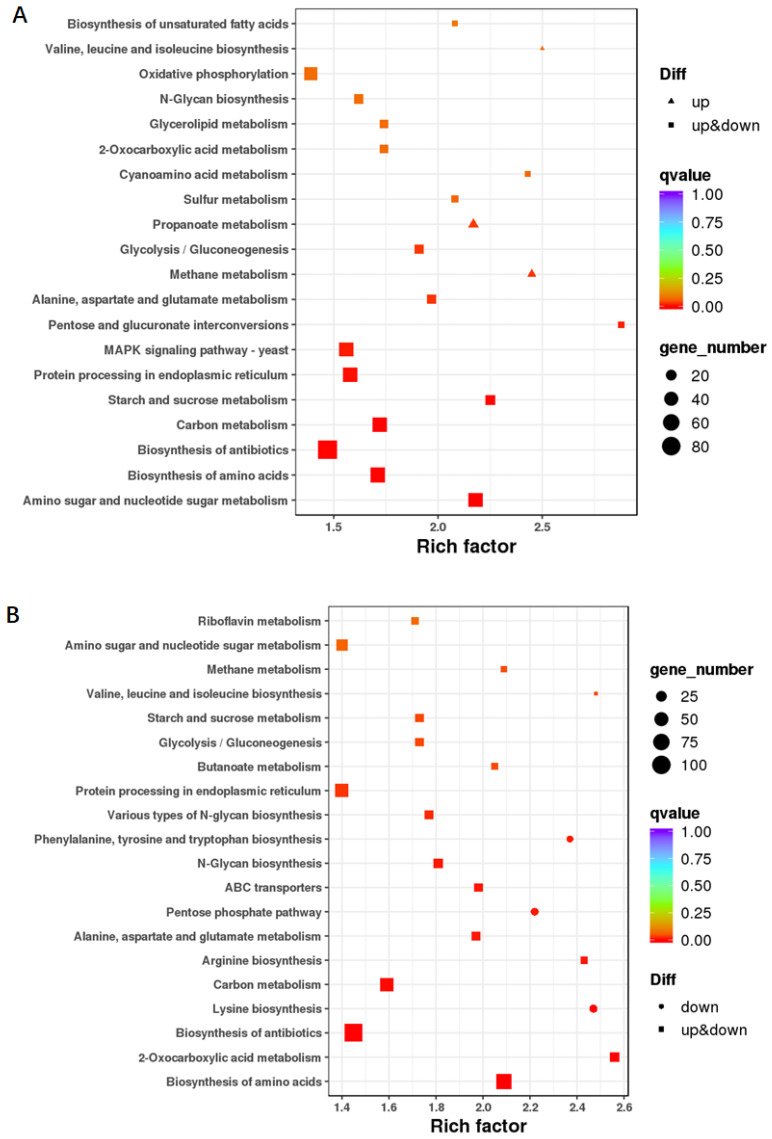
Kyoto Encyclopedia of Genes and Genomes (KEGG) enrichment analyses of differentially expressed genes (DEGs) in C stage vs. pre-C stage (**A**) and C stage vs. post-C stage (**B**). “Rich factor” refers to the ratio of the number of differentially expressed transcripts to the total number of transcripts for each pathway; the larger the rich factor, the higher the degree of enrichment. The q value indicated the P-value after the multiple hypothesis test corrections ranging from 0 to 1; the closer it is to 0, the more significant the enrichment considering a false discovery rate (FDR) ≤ 0.01 as the threshold.

**Figure 9 jof-08-00389-f009:**
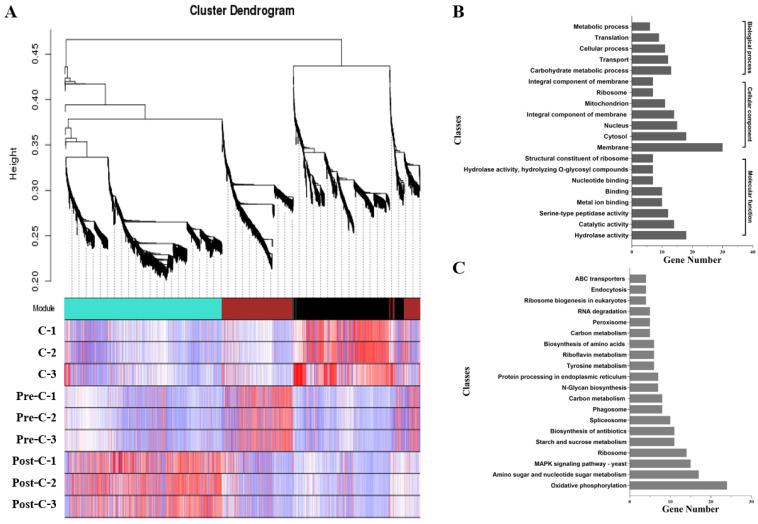
Weighted gene co-expression network analysis (WGCNA) identification of transcriptomes correlated with conidiation. (**A**) Hierarchical cluster tree of DEGs producing three gene co-expression modules; (**B**) The number of DEGs categorized by GO terms in the black module; (**C**) The number of DEGs categorized by KEGG pathways in the black module.

**Figure 10 jof-08-00389-f010:**
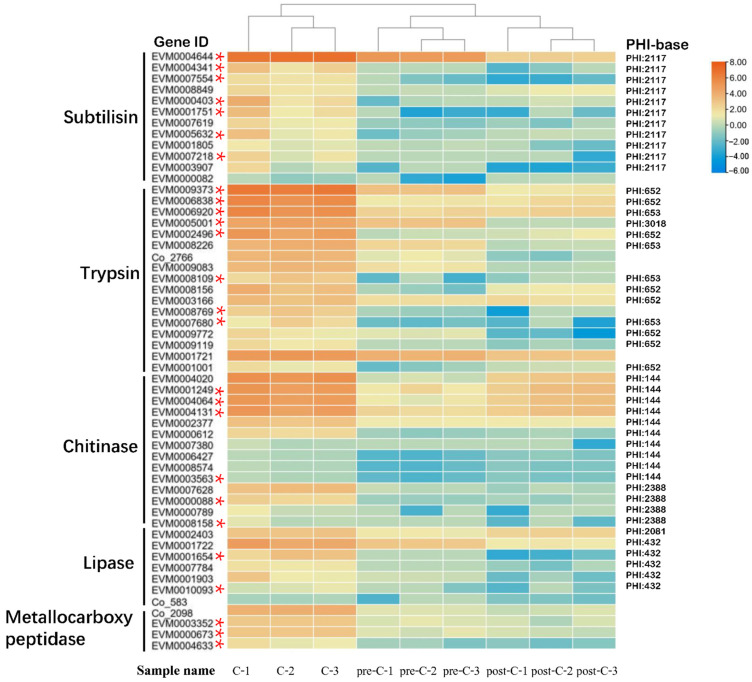
Heatmap of the main DEGs from the black modules of Figure 9 among the nine samples localized into three clusters at different stages (pre-C, C, and post-C). Genes with high and low FPKM values are orange and blue, respectively, whereas * refers to putative secretory proteins.

**Table 1 jof-08-00389-t001:** Genome characteristics of the aphid-obligate pathogen *Conidiobolus obscurus*.

Species	*C. obscurus*
Genome size (Mb)	37.6
Coverage (×)	318
Sequence number	1,145,404
Scaffold number	167
Scaffold N50 length (bp)	1,104,530
Scaffold N90 length (bp)	75,751
G+C content (%)	26.46
Protein-encoding genes	10,262
Average number of exons per genes	4.7
Repeat sequences (bp)	6,558,671
tRNA number, family number	526, 45
rRNA number, family number	8, 2
Other ncRNA number, family number	49, 20

**Table 2 jof-08-00389-t002:** Top ten up-regulated genes in the three conidiation stages.

Internal ID	Pfam Annotation	FPKM ^a^		FC ^b^
		C	Pre-C or Post-C	
**Up-regulated in C vs. pre-C**				
EVM0006541	Polysaccharide deacetylase	220.03	0.06	9.69
EVM0001654	Lipase (class 3)	19.39	0	9.07
EVM0006408	Cell surface GPI-anchored protein	277.16	0.47	8.94
EVM0003421	Cytochrome P450	12.09	0	8.77
EVM0008805	Glycosyl hydrolases family 16	10.70	0	8.66
EVM0002345	TspO/MBR family	291.43	1.14	7.82
EVM0004341	Subtilase family	14.41	0	7.80
Co _242	Homeobox domain	18.18	0	7.70
EVM0009119	Trypsin	5.80	0	7.41
EVM0008624	AhpC/TSA antioxidant enzyme	24.49	0.13	7.09
**Up-regulated in C vs. post-C**				
EVM0009712	TspO/MBR family	289.08	0.04	11.57
EVM0002345	TspO/MBR family	291.43	0.05	11.40
EVM0003678	TspO/MBR family	411.48	0.20	10.92
EVM0007081	TspO/MBR family	305.94	0.14	10.73
EVM0005679	TspO/MBR family	295.98	0.18	10.42
EVM0001538	TspO/MBR family	783.86	0.71	10.21
EVM0001670	Kunitz/Bovine pancreatic trypsin inhibitor	28.37	0	9.65
EVM0010258	Transcription elongation factor SPT6	442.33	0.62	9.59
EVM0003314	Glycosyl hydrolases family 16	21.02	0	9.52
EVM0009807	Kunitz/Bovine pancreatic trypsin inhibitor	19.57	0	9.34

^a^ The transcript level is expressed as fragments per kilobase per million mapped reads (FPKM). ^b^ Fold change (FC) of DEGs based on a FC of ≥2 and an FDR of <0.01.

## Data Availability

Illumina sequence data have been submitted to CNGBdb database under the accession number CNP0002155. All data generated or analyzed during this study are included in this published article, accessions and its Appendix A.

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
