# Peer review of "Genome-Wide Study of Conidiation-Related Genes in the Aphid-Obligate Fungal Pathogen Conidiobolus obscurus (Entomophthoromycotina)"

_jof, 2022, doi:10.3390/jof8040389_

Round 1

Reviewer 1 Report

The Manuscript entitled (Genome-wide study of conidiation-related genes in the aphidobligate fungal pathogen Conidiobolus obscurus (Entomophthoromycotina)) introduced good features in the genome and transcriptome of Conidiobolus obscurus. Also, it is well written and introduced with clear figures and tables but needs some modification.

  • Line 82, Abbreviate Genus name
  • Line 86. (improve their application in agroforestry pest control). Please indicate how this improving will be happened from these results.
  • Figure 5: it needs to be with high resolution to be clear and feasible
  • Add Figure 7 a,b before line 279
  • Figure 9. Transfer it after line 332
  • Figure 10. Transfer it with details in result section. Then, discuss it in discussion section

Author Response

Reviewer 1:

The Manuscript entitled (Genome-wide study of conidiation-related genes in the aphidobligate fungal pathogen Conidiobolus obscurus (Entomophthoromycotina)) introduced good features in the genome and transcriptome of Conidiobolus obscurus. Also, it is well written and introduced with clear figures and tables but needs some modification.

Author response: Thanks for your suggestion to improve our manuscript.

  • Line 82, Abbreviate Genus name

Author response: modified.

  • Line 86. (improve their application in agroforestry pest control). Please indicate how this improving will be happened from these results.

Author response: exploring conidiation-related genes favors manufacturing conidia-based formulation for pest control. And exploring the functional genes related to fungal virulence favors using the pathogen against aphids. I modified this sentence to make it more informative.

  • Figure 5: it needs to be with high resolution to be clear and feasible

Author response: improved.

  • Add Figure 7 a,b before line 279

Author response: transferred.

  • Figure 9. Transfer it after line 332

Author response: transferred.

  • Figure 10. Transfer it with details in result section. Then, discuss it in discussion section

Author response: modified. I also change the figure number due to the genome description before the transcriptome.

Reviewer 2 Report

The manuscript “Genome-wide study of conidiation-related genes in the aphid-obligate fungal pathogen Conidiobolus obscurus (Entomophthoromycotina) employs both genomics and transcriptomics to find candidates genes involved in conidiation in C. obscurus. The manuscript is well structured and seems scientifically sounds. It provides clues for genes involved in conidiation, which are expected to be functionally characterized in future studies to confirm the proposed roles, but also identifies many other genes involved in different functions.

Major comment:

In the Introduction section (line 47), it reads “The fungal entomopathogens secrete subtilisin-like serine proteases to degrade chitin-associated proteins in the insect procuticle”, and then a detailed analysis of these proteins is done in both Results and Discussion sections. It is ok, but the most external layer of the insect cuticle is the epicuticle, rich in lipids and mostly hydrocarbons, which must be assimilated before the fungus reach the procuticle. The main enzymes for hydrocarbon assimilation are cytochrome P450s, and some of them are reported in this study in a general way. P450s also participate in other functions, for example as members of the gene clusters involved in biosynthesis of fungal secondary metabolites. Thus, it would be nice to  include also a characterization of the P450-encoded genes (CYP) found in this study, at least those involved in alkane assimilation (CYP52 family) already characterized in Beauveria bassiana and Metathizium anisopliae.

Minor comments:

  • Line 137: What other contamination besides DNA is expected to be detected by agarose gel electrophoresis?
  • Line 162: Define WGCNA here.
  • Line 306: Please be carefull with this statement and moderate it, no functional analyses (e.g. knockout mutants) were done to affirm “key conidiation genes”

Author Response

Reviewer 2:

The manuscript “Genome-wide study of conidiation-related genes in the aphid-obligate fungal pathogen Conidiobolus obscurus (Entomophthoromycotina) employs both genomics and transcriptomics to find candidates genes involved in conidiation in C. obscurus. The manuscript is well structured and seems scientifically sounds. It provides clues for genes involved in conidiation, which are expected to be functionally characterized in future studies to confirm the proposed roles, but also identifies many other genes involved in different functions.

Major comment:

In the Introduction section (line 47), it reads “The fungal entomopathogens secrete subtilisin-like serine proteases to degrade chitin-associated proteins in the insect procuticle”, and then a detailed analysis of these proteins is done in both Results and Discussion sections. It is ok, but the most external layer of the insect cuticle is the epicuticle, rich in lipids and mostly hydrocarbons, which must be assimilated before the fungus reach the procuticle. The main enzymes for hydrocarbon assimilation are cytochrome P450s, and some of them are reported in this study in a general way. P450s also participate in other functions, for example as members of the gene clusters involved in biosynthesis of fungal secondary metabolites. Thus, it would be nice to include also a characterization of the P450-encoded genes (CYP) found in this study, at least those involved in alkane assimilation (CYP52 family) already characterized in Beauveria bassiana and Metathizium anisopliae.

Author response: Thanks for your suggestions. I added the C. obscurus P450s information in the results.

Minor comments:

  • Line 137: What other contamination besides DNA is expected to be detected by agarose gel electrophoresis?

Author response: for avoiding contamination, nanodrop2000 was used for primarily checking before sample sending to the sequencing platform (Biomarker Co. Beijing, China). To ensure the use of qualified samples for transcriptome sequencing, RNA degradation and contamination, especially DNA contamination, was monitored on 1% agarose gels; RNA purity was checked using the NanoPhotometer spectrophotometer (IMPLEN, CA, USA); RNA concentration was measured using Qubit RNA Assay Kit in Qubit2.0 Flurometer (Life Technologies, CA, USA); RNA integrity was assessed using the RNA Nano 6000 Assay Kit of the Agilent Bioanalyzer 2100 system (Agilent Technologies, CA, USA). This part was modified.

  • Line 162: Define WGCNA here.

Author response: added. Thanks!

  • Line 306: Please be carefull with this statement and moderate it, no functional analyses (e.g. knockout mutants) were done to affirm “key conidiation genes”

Author response: modified, delete the word “key”.